# Free Amino Acids and Biogenic Amines in Canned European Eels: Influence of Processing Step, Filling Medium and Storage Time

**DOI:** 10.3390/foods9101377

**Published:** 2020-09-29

**Authors:** Lucía Gómez-Limia, Roxana Cutillas, Javier Carballo, Inmaculada Franco, Sidonia Martínez

**Affiliations:** 1Food Technology, Faculty of Science, University Campus as Lagoas s/n, University of Vigo, 32004 Ourense, Spain; lugomez@uvigo.es (L.G.-L.); rcutillas@uvigo.es (R.C.); carbatec@uvigo.es (J.C.); inmatec@uvigo.es (I.F.); 2CITACA-Agri-Food Research and Transfer Cluster, Campus Auga, University of Vigo, 32004 Ourense, Spain

**Keywords:** free amino acids, biogenic amines, canning, filling medium, European eels

## Abstract

This study evaluated the effects of the canning process and different filling media on the free amino acid and biogenic amine contents of eels. The main free amino acids were histidine, taurine and arginine, which constituted 72% of the free amino acids in raw eels. All steps in the canning process significantly altered the free amino acid content of eels, relative to raw samples. The changes were influenced by the step, the composition of the frying or filling medium and the storage time. The biogenic amine contents were very low in all samples. Histamine was not detected in either raw eels or canned eels. The highest values were obtained for 2-phenylethylamine. The step of the canning process, the composition of the frying or filling medium and storage time also determined the changes in the biogenic amine contents. The biogenic amines indices were low, indicating the good quality of canned eels.

## 1. Introduction

Fish is a good source of protein, rich in essential amino acids, micro- and macroelements, lipids rich in unsaturated fatty acids and fat-soluble vitamins. However, it is extremely perishable due to the high contents of water, protein and non-protein nitrogen compounds and to the activities of autolytic enzymes, which cause spoilage.

Free amino acids (FAAs) represent one of the most important fractions of non-protein nitrogen in fish. FFAs play a very important role in bacterial spoilage of fish. They undergo important changes during processing and storage, and therefore the different technologies used greatly influence the amino acid profiles. On the other hand, FAAs play an important role in the sweetness, sourness, bitterness and umami taste of foods. In addition, they can interact with reducing sugars (Maillard reaction or non-enzymatic browning), giving rise to different tastes, colors and aromas [1,2]. The FAA content of food and loss of FAA due to processing and storage are also of interest in relation to nutritional aspects. Some amino acids, such as taurine, which is present in high amounts in fish, can have beneficial effects on health [3].

Once the fish is dead, the quality of the flesh degrades rapidly due to chemical reactions (changes in protein and lipid fractions, formation of biogenic amines and hypoxanthine) and microbiological spoilage. Biogenic amines are nitrogenous, low molecular mass organic bases. They are found in a variety of foods due to microbial decarboxylation of the corresponding amino acids or to transamination of aldehydes and ketones by amino acid transaminases [4,5]. Biogenic amine formation depends on factors such as the FAA content, development of microbial activity, processing and preservation conditions during the pre- and post-mortem period [6,7]. However, storage temperature is the most important factor contributing to the formation of biogenic amines [7]. The most important biogenic amines in fish are histamine, tyramine, cadaverine and putrescine, which are formed by the enzymatic decarboxylation of histidine, tyrosine, lysine and ornithine, respectively [8]. Biogenic amines can have negative effects on health. Histamine is important in fish intoxication and is considered an indicator of fish spoilage. Consumption of fish containing high levels of histamine can produce a tingling, burning sensation in the mouth, urticaria, flushing, vomiting and diarrhea [9]. Histamine, tyramine, 2-phenylethylamine and tryptamine can also affect the nervous and vascular systems [10]. Putrescine and cadaverine potentiate histamine toxicity [11]. Some biogenic amines such as putrescine, cadaverine, spermidine and spermine are also considered precursors of carcinogens. Cadaverine has also been found to be a useful indicator of the initial stage of fish decomposition. In canned fish, biogenic amines are used to indicate the freshness of the fish prior to processing and the sanitary conditions during the canning process. Biogenic amines are closely correlated with sensory alterations in fish [12,13]. The profile and content of biogenic amines vary in different fish species [2].

Canning has long been used as a method of preserving fish. However, owing to the thermal sensitivity of some chemical constituents, different breakdown and hydrolytic reactions occur in canned fish. The associated changes during the different steps may be desirable or undesirable, and they can cause important modifications in the nutritional, sensory and safety quality of the final product [9,12,14]. Each step of the canning process contributes to the resulting quality of the final product. The major steps involved in canning fish are pre-cooking, packing with a filling medium in hermetically sealed cans and sterilizing to reach commercial sterility. After these steps, a maturation process begins on storage and continues until the cans reach the consumer.

Various types of filling medium are used for canned fish, including brine and oil. Although vegetable oils are widely used for canning fish, available information on the effects is very limited.

The European eel (*Anguilla anguilla*) is a catadromous species, i.e., it spends most of its life in rivers, during which it grows and matures before migrating to the sea to spawn. Eels are commercially very valuable in Europe (mainly Spain, Portugal, Italy and Netherlands) and Asia (mainly Japan, China, Korea and Taiwan). The female can reach a maximum weight of 9 kg, while the males are smaller, reaching a maximum weight of 2.8 kg, although the specimens in greatest demand are much smaller (<200 g). The eel population has fallen drastically due to various factors threatening its survival, including an increase in contamination of human origin and environmental impacts associated with the construction of diverse obstacles in rivers, such as hydroelectric dams [15]. Canning enables eels to be consumed throughout the year while respecting the state limitations aimed at protecting the species. The use of larger eels for canning (less valued as a fresh product) could also help to increase the reproductive success of the species.

The development of new canned products is also of interest due to the considerable economic importance of such products, their extensive acceptance by consumers, ease of transport and export and nutrient supply.

Although a large number of papers published in recent years have addressed the amino acid and biogenic amine contents of food, information about the loss of free amino acids and formation of biogenic amines during canning and the effects of different steps and different filling media on these components is scarce. In addition, existing studies are limited to a few species, such as tuna, mackerel and sardines, with no reports considering eels.

The present research was undertaken to study the effects of the different steps during canning (frying, sterilization and storage for 2 and 12 months) and of different filling media (sunflower oil, olive oil and spiced olive oil) on the free amino acid content and the formation of biogenic amines in European eels.

## 2. Materials and Methods

### 2.1. Selection and Preparation of Samples

The European eels included in the study were caught in the River Ulla (Galicia, NW Spain) and purchased at a local market (“Plaza de Abastos, Mariscos vivos del Grove”) in Ourense (Galicia, Northwest Spain).

All of the eels used in this study weighed between 200 and 600 g. The eels were eviscerated and transferred to the laboratory, where they were frozen (−20 °C) until canning.

Some randomly selected samples of the frozen eels, hereafter referred to as “raw eels”, were thawed in a refrigerator at +4 °C for 12 h and processed as control samples. The other samples were thawed in 12% brine at room temperature for 45 min before being cut into slices (1.5–2 cm). All slices were mixed to ensure a homogeneous product. The fish slices were fried for 2 min at 190 °C in a deep fryer, to eliminate the water present in the eel and to prevent formation of a water–oil mixture in the cans. Two different frying media were used: sunflower oil (refined sunflower oil) and olive oil (refined olive oil + virgin olive oil). The fried slices of eel were cooled (30 °C) and placed in cans (6 or 7 slices in each). The hot filling medium was then added: sunflower oil (eels previously fried in sunflower oil), olive oil or olive oil plus chili and pepper (eels previously fried in olive oil). The olive oil plus chili and pepper filling medium will be referred to hereafter as “spiced olive oil”. The cans were then vacuum-sealed and sterilized. The time/temperature combination used for sterilization was 118 °C and 30 min (F0 = 11). Finally, the cans were cooled and stored at room temperature. The gross weight of each can was 185.68 ± 4.29 g and the drained weight 50.00 ± 3.91 g.

Eels were sampled raw (control), after each processing step (frying and sterilization treatment) and at two different times throughout the storage of each final product (2 and 12 months of room storage). For biogenic amine determinations, the eels were also sampled after thawing-salting, as biogenic amines can be formed during this step.

### 2.2. Moisture and pH Determination

The moisture content was determined after drying the eels to constant weight in an oven for 16 h at 105 ± 1 °C [16]. The pH was measured using a digital pH meter (Crison, model GLP21, Barcelona, Spain) according to the AOAC (Association of Official Analytical Chemists) methods [16].

### 2.3. Free Amino Acid (FAA) Determination

The FAAs were extracted and derivatized following the procedure described by Franco et al. [17], with some modifications. Fish samples (2 g) were thoroughly homogenized in 20 mL of 0.6 N HClO_4_ in a lab blender (Ultra-Turrax^®^ T25, IKA, Staufen, Germany) for 2 min. The samples were then centrifuged at 1800× *g* for 20 min. The supernatant was collected and then filtered. The pH of the filtrate was adjusted to 7.1 ± 0.2 with 30% KOH and then cooled for 10 to 20 min until reaching a temperature of 2 °C.

For derivatization, 0.2 mL of standard solution or hydrolyzed sample was dried in a vacuum centrifuge concentrator at 37 °C. A 20 µL aliquot of the derivatizing solution (ethanol + Milli-Q water + triethylamine + phenyl isothiocyanate) was then added to the samples, and the solution was mixed and left to stand at room temperature for 20 min. The resulting solution was evaporated in a vacuum centrifuge concentrator at 37 °C. The dry residue was resuspended in 500 µL of diluent solution disodium acid phosphate (675.4 mg) + Milli-Q water (950 mL) and filtered (through Waters 0.45 µm pore diameter filters).

The FAAs were identified and quantified by HPLC, under the conditions described by Franco et al. [17], with some minor modifications. The liquid chromatography equipment consisted of a chromatograph (ThermoFinnigan, Silicon Valley, CA, USA) with a UV/VISIBLE photodiode array detector (Spectrasystem UV6000LP). The samples were separated on a reversed phase column of diameter 4.6 mm and length 25 cm (C18 Ultrasphere 5–ODS, from Beckman, Fullerton, CA, USA). The temperature of the column was maintained at 50 ± 1 °C with a column heater (Spectrasystem 3000). The wavelength of the detector was 254 nm. Standards of the 19 different amino acids were supplied by Sigma Chemical Co. (St Louis, MO, USA). Data regarding free amino acid composition were expressed in mg/100 g of muscle. All the samples and standards were injected into the column at least in duplicate.

Repeatability tests were performed by injecting a standard and a sample into the column consecutively six times in a day. Reproducibility tests were also carried out by injecting the standard and the sample into the column twice a day for three days under the same experimental conditions. The results obtained in these tests were not significantly different (*p* < 0.05).

### 2.4. Biogenic Amine Analysis

Biogenic amines were analyzed by the method described by Lorenzo et al. [18]. Fish muscle (5 g) was mixed with 10 mL of 0.6 N HCLO_4_ and 1 mL of internal standard (1–7 diaminoheptane). The mixture was homogenized with a lab blender (Ultraturrax) for 2 min and then centrifuged at 3000 rpm for 10 min at 4 °C. The supernatant was collected, and the same process was repeated with the residue for complete extraction. Finally, the two supernatants were placed in a 25 mL volumetric flask and 0.6 N HClO_4_ was added to make up the final volume.

For derivatization, an aliquot (0.5 mL) of each extracted sample or of the standard solution of any biogenic amine was immediately placed in a tube, and 100 μL of 2 N NaOH, 150 μL of a saturated solution of NaHCO_3_ and 1 mL of dansyl chloride were added consecutively. The tube was shaken gently and placed in a water bath at 40 °C for 45 min. In order to remove dansyl chloride residue, 50 μL of ammonia was then added and the mixture was left to stand for 30 min. Finally, the volume was made up to 2.5 mL with acetonitrile and the mixture was filtered through 0.25 μm pore-size filters prior to HPLC analysis.

Separation, identification and quantification of the biogenic amines were carried out by HPLC, following the procedure described by Eerola et al. [19], with the aforementioned HPLC equipment. The different biogenic amines were separated on a reversed phase column of diameter 4.6 mm and length 25 cm (C18 Ultrasphere 5–ODS, from Beckman, Fullerton, CA, USA). The temperature of the column was 40 ± 1 °C and the wavelength of the detector 254 nm.

Separation was achieved at a flow rate of 1 mL/min, with a gradient between two solvents: a solution of 0.1 M ammonium acetate was used as eluent A and acetonitrile as eluent B.

A standard solution containing appropriate amounts of tryptamine, 2-phenylethylamine, putrescine, cadaverine, histamine, tyramine, spermidine and 1,7-diaminoheptane (as an internal standard) was used to quantify the biogenic amines present in the samples. Each biogenic amine was expressed in mg/kg.

Repeatability and reproducibility tests were also performed in the biogenic amine analysis. The results of the tests were not significantly different (*p* < 0.05).

The biogenic amine index (BAI) was estimated using the equation described by Veciana-Nogués et al. [13]:Biogenic amine index = (histamine + cadaverine + putrescine + tyramine)(1)

### 2.5. Statistical Analysis

All analyses were carried out at least in triplicate. The data were examined by one-way analysis of variance (ANOVA), and the least squares test (LSD) was used (*p* < 0.05) to compare the mean values. The tests were implemented using Statistica software version 8.0 (Statsoft © Inc., Tulsa, OK, USA). Canonical discriminant analysis (CDA) was used to classify the eel samples. The CDA variables were selected by principal components extraction and linear discriminant analysis, and the variables with the highest discriminatory capacity were selected to establish which FAAs and biogenic amines can be used to discriminate and classify eels according to the canning step and type of filling medium.

## 3. Results and Discussion

### 3.1. Moisture and pH Values

The moisture contents of raw eels and eels packed in sunflower oil, olive oil and spiced olive oil, at each step of the canning process and after room storage (for 2 and 12 months), are shown in Figure 1A. The moisture content of raw eel was 74.15 ± 0.64%. Frying in sunflower oil and in olive oil resulted in water content losses of 11.14 and 15.75%, respectively. Sterilization treatment did not cause changes in the moisture content of canned eels packed in sunflower oil or in olive oil. However, sterilization caused a decrease in the moisture content of canned eels packed in spiced olive oil.

After storage for two months, the moisture content of the eels decreased and then remained stable for up to twelve months of storage. The eels packed in olive oil had a lower moisture content (43.21%) than canned eels packed in sunflower oil and in spiced olive oil. Water can be lost due to the thermal treatment and to denaturation of proteins in muscle, which causes a decrease in the water holding capacity of the myofibrillar protein fraction [14]. In addition, frying, sterilization and storage cause an interchange between the water of the fish and filling medium, causing an increase in lipid content and a decrease in moisture content. In the canned eels packed in spiced olive oil, some active components could also pass into the muscle.

The pH can significantly affect protease activity and production of biogenic amines. The pH of raw eels was high, between 6.25 and 6.31 (Figure 1B). The pH increased significantly during frying (6.48 and 6.60 in eels fried in sunflower oil and olive oil, respectively). Sterilization caused a further increase in the pH of canned eels packed in sunflower oil and in spiced olive oil; however, it did not vary in canned eels in olive oil. The increases can be attributed to proteolysis, breakdown of protein and enzymatic activity. During storage, the pH remained stable at the beginning of storage and decreased after 12 months of storage in canned eels packed in sunflower oil and in spiced olive oil. In canned eels packed in olive oil, the pH decreased throughout storage. This decrease may be due to the degradation of lipids and other components during storage. The change in pH varied depending on the filling medium. The pH was lowest in canned eels packed in olive oil.

Several studies have reported the relationships between pH and biogenic amine accumulation. A pH between 4 and 5.5 favors decarboxylase activity [10]. In this study, significant (*p* < 0.05) positive correlations between pH and putrescine (*r* = 0.75), between pH and cadaverine (*r* = 0.45) and between pH and spermidine (*r* = 0.48) were observed.

### 3.2. Free Amino Acids

The FAA content is very important in order to evaluate protein-rich food. The FAA contents of the muscle tissue of the raw eel and of eels at each step of the canning process and after room storage (for 2 and 12 months) are presented in Table 1 and Table 2.

The main FAAs in raw eels were histidine (222.76 ± 6.99 mg/100 g), taurine (91.63 ± 6.62 mg/100 g) and arginine (80.58 ± 6.07 mg/100 g), which constituted 72% of the total FAA content.

Histidine accounts for 41.11% of the total FAAs. This result is consistent with those of previous studies [1,20]. The histidine contents of fish muscles can vary significantly due to differences in fish species, sex, season, feeding, living environment, swimming activity and stage of maturity [21]. Histidine is important because of its physiological and nutritional roles in fish. Some migratory species such as tuna, skipjack and mackerel contain high amounts of histidine that maintain the muscle pH level during swimming and that act as an energy source during prolonged starvation [21]. The eel is also a migratory species. This may be the reason for the high histidine content in the eels analyzed, although it is lower than in other species such as tuna and mackerel [20]. Antoine et al. [20] reported important variations in histidine content of individual fish of the same species and between red and white muscle. On the other hand, histidine is the precursor of histamine, which mainly appears when extreme temperatures are used.

Taurine represented 16.92% of total FAAs in raw eels. The values obtained are similar to those reported for megrim (95 mg/100 g) by Gormley, Neumann, Fagan and Brunton [22]. These authors reported a wide range of taurine content in 14 fish species (between 7 and 176 mg/100 g). Taurine is a derivative of cysteine. It is not strictly an amino acid, and as it contains a sulfonic acid group, it is referred to as sulfonic acid. Some fish such as mackerel, seabream and tuna have a high proportion of taurine in their muscle tissue [21,23]. Although taurine does not have an important impact on taste, this free amino acid is known to benefit human health due to anti-inflammatory, anti-atherosclerotic effect and anti-obesity effects [3].

Arginine accounts for 14.87% of the FAA content of raw eels. Arginine has anti-aging and anti-fatigue effects, but it is a precursor of putrescine. Arginine and histidine are associated with a bitter taste [1].

Glycine was detected in relatively high amounts in the raw eels (50.24 ± 2.29 mg/100 g). This amino acid is an indicator, together with hydroxyproline, of the presence of connective tissue, principally collagen, and plays a key role in its stability [24]. Glycine is associated with a sweet taste [1].

Glutamic acid (21.60 ± 1.75 mg/100 g) and aspartic acid (14.24 ± 0.54 mg/100 g) were also quantitatively important. They play an important role in the umami taste of food.

Other amino acids such as alanine, serine and threonine, which are associated with a sweet taste, were also found in appreciable quantities (12.37, 3.81 and 9.07 mg/100 g, respectively). Valine, isoleucine, leucine, phenylalanine and tryptophan contents of raw eels were 1.73, 2.47, 4.08, 4.83 and 5.24 mg/100 g, respectively. These amino acids contribute a bitter taste.

The other amino acids were found in lower quantities in raw eels.

The non-essential amino acid (NEAA) content was 286.31 ± 19.25 mg/100 g in raw samples.

The essential amino acid (EAA) content is a major factor that affects the nutritional value of protein. In raw eels, the EAA content was 255.53 ± 6.59 mg/100 g. The value of the essential/non-essential (EAA/NEAA) ratio for FAAs was 0.90 ± 0.08.

The FAA composition can vary significantly according to the production conditions during the preservation and processing of the food. Significantly different (*p* < 0.05) quantities of some FAAs were detected at all steps of the canning process.

Frying greatly altered the FAA content of eels, relative to raw samples. The changes were influenced by the composition of the frying oil and were not homogeneous for the different FAAs, because the concentrations of some amino acids decreased, while those of others increased. In the eels fried in sunflower oil (Table 1), significant increases (*p* < 0.05) in serine, proline, tyrosine, taurine, histidine, valine and lysine contents and a decrease in the ornithine content were observed. Total essential and non-essential free amino acids increased during frying in sunflower oil.

In eels fried in olive oil (Table 2), the EAA and NEAA contents both increased significantly. Arginine, alanine, proline, tyrosine, taurine, valine, isoleucine, leucine, phenylalanine, lysine and tryptophan contents increased during frying in olive oil, while glycine and ornithine contents decreased.

The increase in FAAs during frying may be derived from the moisture lost during process. However, some FAAs increased and other decreased, suggesting that moisture was not the only factor affecting the FAA content during the frying process. Some changes can be attributed to the heat-induced denaturalization of protein, to the decomposition of FAAs and to the formation of different volatile compounds such as pyrazines and sulfides, which have important impacts on the aroma and flavor of fried fish [25]. On the other hand, amino acids can react with sugar, giving rise to the Maillard reaction. The changes during frying can cause variations in fish flavor, especially those associated with sweet and bitter tastes, as the amino acids associated with these parameters underwent the most notable changes.

In general, the sterilization process also had a significant effect on the FAA composition of the European eel samples. The changes were also influenced by the composition of the filling medium and were not homogenous for the different free amino acids. In canned eel packed in sunflower oil (Table 1), aspartic acid, arginine, proline, valine, leucine and tryptophan contents increased significantly after the sterilization process (*p* < 0.05). However, glutamic acid, serine, glycine, taurine and histidine contents decreased. Loss of glutamic acid is of great interest because, although glutamic acid is a NEAA, it is an important source of nitrogen and it is involved in taste perception, contributing to the umami taste [26]. Both total EAA and NEAA contents were lower in canned eels packed in sunflower oil than in those packed in the other media.

In canned eels packed in olive oil (Table 2), significant increases (*p* < 0.05) in aspartic acid, glycine, arginine, ornithine, histidine, threonine and valine, and decreases in serine and phenylalanine, were observed after the sterilization process. In the canned eels packed in spiced olive oil, aspartic acid, glycine, arginine, alanine, histidine, threonine and valine contents increased, while serine and proline contents decreased after the sterilization process.

Therefore, important variations between canned eels packed in different filling media were observed. This may be due to different interactions between FAAs and filling oil. There were also differences between canned eels packed in olive oil and canned eels packed in spiced olive oil, and therefore the spices influenced the changes in amino acids during sterilization.

Aubourg [14] reported that FAAs could be lost as a result of extraction by the filling medium and/or interaction reactions with oxidized lipids. The different fatty acid compositions of the filling media also condition the heat penetration, which also conditions the effect of the treatments on the different components [27]. On the other hand, the different biogenic amines levels in canned eels could be due to the different degrees of heat-induced degradation of FAA in different filling media.

The EAA/NEAA ratio is considered a good way of estimating the production of free essential amino acids during processing. However, in canned eels, this ratio did not vary during the sterilization process, as the contents of both total FAAs and total NEAAs increased.

Some studies have reported changes in individual amino acids caused by heating [14]. Heat processing causes denaturalization of protein. The denatured proteins are more reactive and can interact with other constituents. Other FAAs form amines, volatile acids and other nitrogenous substances.

Canned storage is normally necessary to produce satisfactory textural and optimal palatability of canned fish [14]. During storage, many compounds migrate from the fish to the filling medium and vice versa, in a dynamic equilibrium influenced by the characteristics of the fish muscle and the filling medium as well as by the type of processing and storage. The values reported in Table 1 and Table 2 show that the FAAs underwent greater changes during storage. The changes were also influenced by the composition of the filling medium and storage time. In canned eels packed in sunflower oil, storage for 2 months caused increases in glutamic acid, hydroxyproline, glycine, arginine and proline and decreases in serine, valine and leucine.

In canned eels packed in olive oil, storage for 2 months led to loss of aspartic acid, glycine, tyrosine, taurine, valine and leucine and an increase in the arginine content. In canned eels packed in spiced olive oil, the losses were lower during 2 months of storage. Only the arginine content decreased, and the serine, proline, isoleucine and leucine contents increased.

The greatest changes occurred after 12 months of storage. In canned eels packed in sunflower oil, the hydroxyproline, serine, arginine, alanine, proline, tyrosine, histidine, valine, leucine, lysine and tryptophan contents increased, and the aspartic acid and glycine contents decreased after storage for 12 months.

In canned eels packed in olive oil and spiced olive oil, hydroxyproline, arginine and valine contents increased, and glycine, histidine and lysine contents decreased. In the canned eels packed in spiced olive oil, aspartic acid, glutamic acid, serine, taurine, leucine and phenylalanine contents also decreased after 12 months of storage. In canned eels packed in olive oil and in spiced olive oil, there was a significant decrease in EAA and an increase in NEAA and a consequent significant decrease in the EAA/NEAA ratio.

These results suggest that the hydrolysis reactions and interactions continued during room storage. On the other hand, the exchange of FAAs and the interactions appear to differ depending on the filling medium.

### 3.3. Biogenic Amine Composition

Biogenic amines are found in very low levels in fresh fish, and their formation is associated with bacterial spoilage. Exposure of fish muscle to high temperatures causes an increase in biogenic amine formation in fish [7]. The levels of the biogenic amines found in the samples of fish are shown in Table 3. In all samples studied, the biogenic amine contents were generally very low.

Histamine is produced in raw fish due to the action of bacterial histidine decarboxylase at high temperatures and/or prolonged exposure. Thawing of frozen fish and long storage time at room temperature before canning can also lead to accumulation of histamine. Histamine is very heat resistant and it can remain intact during the sterilization process or in other processed fish products [27]. The legal limit for histamine established for fish products by the US Food and Drug Administration [28] is 50 mg/kg, and that established by the European Commission [29] is 100 mg/kg. Although eels have a high histidine content, histamine was not detected in either raw eels or canned eels (<LOD), which showed that the fish used in these products were fresh, and the products were produced using good manufacturing practices. Adequate storage temperatures and duration of refrigeration prevent the formation of histamine [7,28]. Özogul, Özogul and Gökbulut [30] reported 0.5 mg histamine/100 g of muscle of eel stored on ice for 1 day, and the content increased with temperature and time of storage. Veciana-Nogués et al. [13] observed very low histamine content in both fresh and canned tuna. In a study of various different fish products, Zhai et al. [5] reported the maximum histamine levels in canned anchovies (26.95 mg/kg) and canned sardines (22.38 mg/kg) and a histamine content of less than 10 mg/kg in all other canned samples tested. Mohan et al. [27] and Barbosa et al. [12] observed that cooking tuna before canning increased the histamine content.

The 2-phenylethylamine and tyramine have an aromatic structure [31], which indicate a vasoconstrictor activity.

The 2-phenylethylamine content of raw eels was 3.10 ± 0.73 mg/kg. Similar results were reported for eels by Özogul et al. [30]. Salting did not cause changes in 2-phenylethylamine content (3.05 ± 0.72 mg/kg); however, frying increased the content. The increase was similar in eels fried in sunflower oil (4.54 ± 0.25 mg/kg) and in eels fried in olive oil (4.97± 0.30 mg/kg). No significant differences (*p* > 0.05) were observed before and after the sterilization step in eels fried in sunflower oil (4.01 ± 0.21 mg/kg). However, the 2-phenylethylamine content decreased in canned eels packed in olive oil (3.64 ±0.44 mg/kg) and in spiced olive oil (3.02 ± 0.12 mg/kg) after the sterilization process.

The different 2-phenylethylamine levels in canned eels could be due to the different degrees of heat-induced degradation of FAA in different filling media and the different protective effects of the filling media.

Veciana-Nogués et al. [13] observed a slight increase in 2-phenylethylamine content of canned tuna after cooking and after packing, and they did not observe this biogenic amine after the sterilization step. Storage caused a significant increase (*p* < 0.05) in the 2-phenylethylamine content of canned eels, mainly after 12 months of storage in sunflower oil and in spiced olive oil. Generally, 2-phenylethylamine is formed when tyramine concentration is high, and its presence can be related to a non-specific activity of tyrosine decarboxylase [32].

The tyramine content of raw eels was 1.61 ± 0.07 mg/kg. During salting, the tyramine concentrations increased significantly (by 5.69 ± 0.95 mg/kg) (*p* < 0.05). Subsequent frying did not cause significant changes in the tyramine content when sunflower oil was used but caused a decrease when olive oil was used. In canned eels packed in sunflower oil, the sterilization process caused a significant decrease (*p* < 0.05) in the tyramine content. However, the tyramine content did not vary during the sterilization step in canned eels packed in olive oil or in spiced olive oil. During storage, significant decreases were observed in tyramine content, mainly after 12 months of storage. Tyramine was not detected even after 12 months of storage in canned eels in sunflower oil and in spiced olive oil, and it decreased to 0.83 mg/kg in canned eels packed in olive oil. Bilgin and Gençcelep [10] observed that tyramine was detected in canned tuna, chunk canned tuna, marinated anchovies, canned mackerel and canned sardines at levels ranging between ND and 48.63 mg/kg.

The concentration of putrescine ranged between 0.41 and 0.52 mg/kg in raw eels. Özogul et al. [30] reported higher values in eels stored in ice for 1 day (8.6 mg/kg), and the content increased with an increase in the temperature and time of storage. Salting increased the putrescine content (1.89 ± 0.29 mg/kg). However, the subsequent frying and sterilization process decreased putrescine content in both olive oil and sunflower oil. Storage for 12 months caused a significant decrease in putrescine content in all samples. The putrescine levels can decrease because this biogenic amine is an intermediate product in the synthesis of spermidine and spermine [33]. Zarei et al. [4] found that putrescine was detected in different canned tuna samples in the range 0.29–52.83 mg/kg. Zhai et al. [5] reported that putrescine was detected within the range ND-25.01 mg/kg. Bilgin and Gençcelep [10] observed that putrescine was detected in canned sardines, canned mackerel and marinated anchovies in the range ND-57.30 mg/kg.

Cadaverine content is a good indicator of spoilage. The results in Table 3 show that the mean value of cadaverine was 1.40 ± 0.15 mg/kg in raw eels. Cadaverine was formed, especially after the salting process. The increase in cadaverine content after salting can be explained by microbial contamination during handling and salting. Frying and sterilization did not cause changes in cadaverine content. The cadaverine content of canned eels packed in sunflower oil was stable after sterilization up to 2 months of storage and then decreased. However, in the canned eels packed in olive oil, it increased throughout storage. When spices were used in the filling medium, the cadaverine content increased during the first two months of storage but then decreased for up to 12 months. The different behavior of the cadaverine, as with other biogenic amines, may be due to the reactions with the filling medium and the different extraction processes. Prester [9] pointed out that biogenic amine content may decrease due to the loss to the filling medium and elimination prior to analysis of canned muscle. Barbosa et al. [12] observed an increase during cooking and a marked decrease during canning of skipjack tuna.

The spermidine content of raw eels was 2.36 ± 0.17 mg/kg. The content increased during the thawing-salting process. As for other biogenic amines, the increase may be due to handling and mechanical preparation during thawing-salting. Barbosa et al. [12] also observed an increase in spermidine content during thawing of skipjack tuna.

Frying caused a decrease in the spermidine content. The sterilization step did not affect the spermidine content of canned eels packed in sunflower oil or olive oil; however, this step decreased the spermidine content in canned eels packed in spiced olive oil. Veciana-Nogués et al. [13] observed a decrease in spermidine content of canned tuna after cooking and sterilization. Barbosa et al. [12] reported very variable spermidine contents in cooked and canned samples, which depended on the degree of handling. Storage for 2 months did not cause changes in spermidine content in canned eels packed in sunflower oil or olive oil; however, it increased the spermidine content of canned eels in spiced olive oil. Storage for 12 months caused a decrease in the content, mainly in canned eels packed in sunflower oil or spiced olive oil.

Different studies have pointed out that the presence of spermidine and other biogenic amines such as putrescine and cadaverine in canned samples can be considered a health hazard, as biogenic amine in combination with nitrite can lead to the formation of nitrosamines, which are known to be carcinogenic [34].

The presence of biogenic amines is not only important from a health point of view but because these substances can be used as indicators of the degree of freshness of food, as well as spoilage and sensory quality [12,13].

Zhai et al. [5] observed that the total biogenic amine content in different canned fish products ranged from 1.94 to 112.54 mg/kg, with a mean value of 46.43 mg/kg.

As noted above, salting caused an increase in total biogenic amines, probably due to handling during the process. However, during frying and sterilization, a decrease in the biogenic amine content was observed. As biogenic amines are heat resistant, the decrease may be due to a loss to the oil medium. The total biogenic amine content increased during the first two months of storage, before decreasing to 12 months. This may be due to the breakdown of protein, reactions with the filling medium and the different extraction processes. Differences between filling media were observed. The final product with the highest amine content was canned eels in olive oil. Nevertheless, the values obtained for both total biogenic amine content and biogenic amine index are very low, well below recommended levels in all samples. The FDA [28] recommended maximum values of 1000 mg/kg of total biogenic amines in fish. The European Commission [35] recommended that the maximum level of total biogenic amines should be less than 200 mg/kg in fish and fish products.

The biogenic amine index varied in a similar way. All values of biogenic amine index were below 50 mg/kg, indicative of good quality food [13], so that good quality can be concluded in all cases.

However, the quality indices depend on many factors, mainly concerning the nature of the product (species, fresh, salted, heat treatment, storage, canned). Therefore, more studies on biogenic amines are required in order to establish the limit for fish acceptability.

### 3.4. Statistical Analysis

Multivariate statistical techniques were used with the aim of discriminating between eels at different stages of processing and canning with different filling media. The data on all free amino acids and biogenic amines studied were included in the factorial analysis to obtain the variables that contributed most to the classification.

The comparison for free amino acids is shown in Figure 2. Two discriminating functions were statistically significant (*p* < 0.05) in each case. In all cases, the most significant function (*p* < 0.05) was F1. Five groups were clearly separated in canned eels packed in sunflower oil, each corresponding to a specific step during canning (Figure 2A). In the canned eels packed in olive oil (Figure 2B) or in spiced olive oil (Figure 2C), some overlap between fried eels, sterilized eels and canned eels stored for 2 months was observed. These results indicate that the greatest changes occurred during frying and these were lower during sterilization and the first two months of storage. The linear discriminant analysis clearly classified the canned eels stored for 12 months (Figure 2D), indicating that the greatest changes take place at this last stage. In the eels packed in sunflower oil (Figure 2A), the FAAs with the highest discriminatory power were proline, tyrosine, taurine, histidine, threonine, lysine and tryptophan. In the eels packed in olive oil (Figure 2B), hydroxyproline, arginine, tyrosine and lysine were the fatty acids with the highest discriminatory power. In the eels packed in spiced olive oil (Figure 2C), these corresponded to glutamic acid, tyrosine, histidine, isoleucine and tryptophan. In addition, discriminant analysis selected 10 free amino acids (serine, histidine, threonine, lysine, glycine, hydroxyproline, arginine, tyrosine, leucine and proline) able to discriminate the different types of canned eels packed with different filling media after storage for 12 months (Figure 2D).

Discriminant analysis was also used to differentiate raw and canned eels according to biogenic amine content (Figure 3). Two discriminatory functions were statistically significant (*p* < 0.05), indicating the ability of biogenic amines to discriminate the eel samples. Figure 3A shows the classification of eels packed in sunflower oil. Two biogenic amines, phenylalanine and putrescine, showed high discriminatory power. The percentage of correctly classified eels with these variables is 99%. Figure 3B shows the classification of eels packed in olive oil. Putrescine, tyramine and spermidine displayed the highest discriminatory power, with correct classification of 98% of the different eel products. Figure 3C shows the classification of eels packed in spiced olive oil. In this case, putrescine and spermidine also displayed a high level of discriminatory power, with correct classification of 98% of the products.

In the canned eels stored for 12 months (Figure 3D), some overlap between canned eels packed in sunflower oil and canned eels packed in spiced olive oil was observed. Phenylalanine, tyrosine and spermidine displayed a high level of discriminatory power.

## 4. Conclusions

The study findings showed that canned European eels are a good source of free amino acids, with a good balance of essential amino acids, and that they provide very small amounts of biogenic amines.

In general, canning, filling medium and duration of storage of European eels had significant effects on the free amino acid and biogenic amine contents of the fish. The changes varied depending on the free amino acid or biogenic amine considered.

The main free amino acids were histidine, taurine and arginine. Significant differences (*p* < 0.05) in the quantities of some free amino acids were detected at all steps of the canning process. The changes were influenced by the composition of the frying oil and filling medium and continued during room storage.

The biogenic amine contents of all samples were very low. Histamine was not detected in raw eels or in canned eels. The highest values were obtained for 2-phenylethylamine. The addition of spices to canned eels packed in olive oil appeared to reduce the biogenic amine content of the final product, except that of 2-phenylethylamine. Canned eels packed in spiced olive oil had a similar biogenic amine content as the canned eels packed in sunflower oil after 12 months of storage.

The findings provide a greater understanding of the influence of canning and filling medium on free amino acids and biogenic amine composition and can be used to improve the quality of this type of product.

## Figures and Tables

**Figure 1 foods-09-01377-f001:**
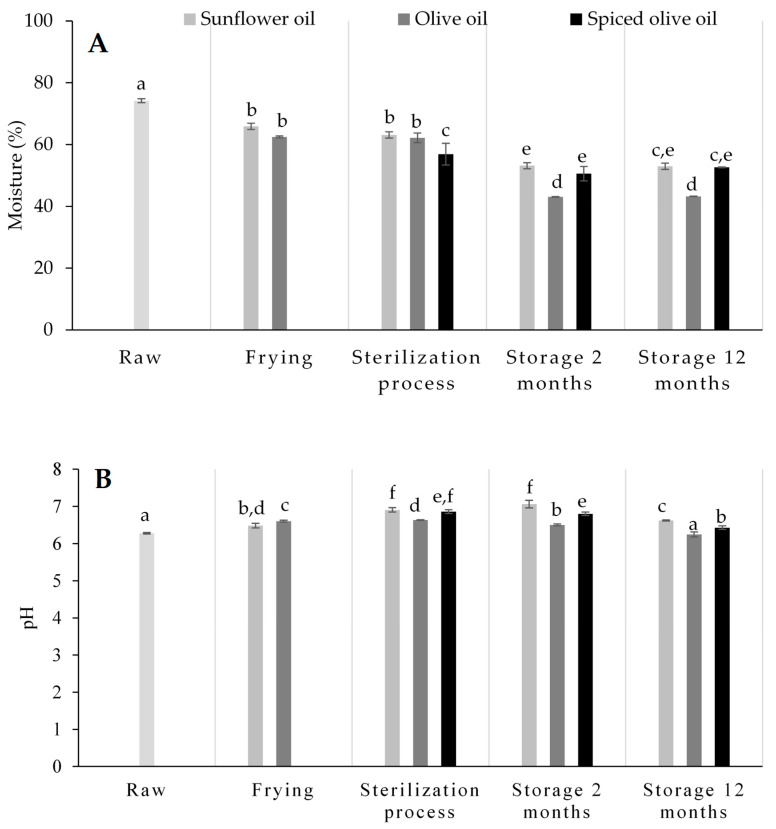
Moisture content (%) (**A**) and pH values (**B**) of raw and canned European eels packed in sunflower oil, olive oil and spiced olive oil throughout the different steps of the canning process and after 2 and 12 months of storage. Values with different superscripts were significantly different (*p* < 0.05).

**Figure 2 foods-09-01377-f002:**
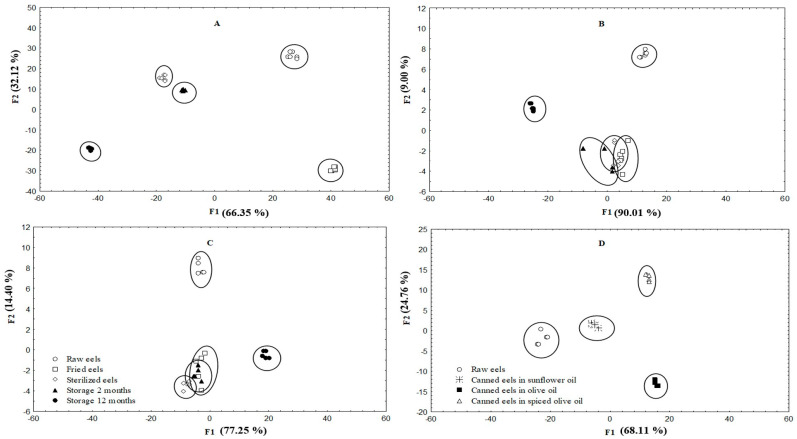
Scatter plot of data obtained in linear discriminant analysis (LDA) for raw and canned European eels packed in sunflower oil (**A**), olive oil (**B**) and spiced olive oil (**C**), at each step of the canning process and after storage for 2 and 12 months. (**D**) represents the plot of the three types of canned eels after storage for 12 months.

**Figure 3 foods-09-01377-f003:**
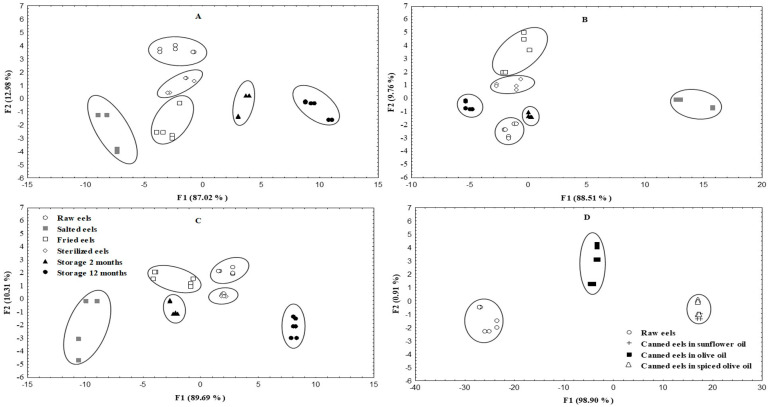
Scatter plot of amine content data obtained in linear discriminant analysis (LDA) for raw and canned European eels packed in sunflower oil (**A**), olive oil (**B**) and spiced olive oil (**C**), at each step of the canning process and after storage for 2 and 12 months. (**D**) represents the plot of the three types of canned eels after storage for 12 months.

**Table 1 foods-09-01377-t001:** Free amino acid content of raw and canned European eels packed in sunflower oil throughout the different steps of the canning process and after 2 and 12 months of storage (expressed mg/100 g muscle).

	Raw	Frying	Sterilization	Storage 2 Months	Storage 12 Months
***NON-ESSENTIAL AMINO ACIDS***					
Aspartic acid	14.24 ± 0.54 ^a^	11.23 ± 2.60 ^a^	26.34 ± 1.39 ^b,c^	28.22 ± 0.39 ^c^	23.64 ± 1.93 ^b^
Glutamic acid	21.60 ± 1.75 ^a,b^	23.11 ± 1.07 ^a^	15.90 ± 0.45 ^c^	20.37 ± 0.84 ^a,b^	19.92 ± 1.09 ^b^
Hydroxyproline	0.86 ± 0.04 ^a,b^	1.02 ± 0.01 ^a^	0.79 ± 0.03 ^b^	1.33 ± 0.14 ^c^	2.13 ± 0.09 ^d^
Serine	3.81 ± 0.03 ^a^	4.67 ± 0.12 ^b^	2.83 ± 0.45 ^c^	1.62 ± 0.02 ^d^	3.04 ± 0.44 ^c^
Glycine	50.24 ± 2.29 ^a^	57.10 ± 2.40 ^a^	39.89 ± 2.22 ^b^	56.12 ± 5.78 ^a^	42.30 ± 4.88 ^b^
Arginine	80.58 ± 6.07 ^a^	76.44 ± 2.02 ^a^	128.16 ± 0.22 ^b^	140.34 ± 5.90 ^c^	159.91 ± 8.90 ^d^
Alanine	12.37 ± 1.13 ^a^	12.20 ± 0.50 ^a^	13.54 ± 1.98 ^a^	15.72 ± 3.65 ^a^	22.40 ± 1.77 ^b^
Proline	2.68 ± 0.28 ^a^	4.77 ± 0.26 ^b^	5.20 ± 0.18 ^b^	6.66 ± 1.23 ^c^	8.09 ± 0.71 ^d^
Tyrosine	5.98 ± 0.32 ^a^	8.97 ± 0.33 ^b^	9.37 ± 0.19 ^b^	10.25 ± 1.62 ^b^	12.21 ± 1.03 ^c^
Taurine	91.63 ± 6.62 ^a^	169.73 ± 0.57 ^b^	65.08 ± 2.93 ^c^	76.21 ± 7.04 ^c^	75.57 ± 6.88 ^c^
Ornithine	1.58 ± 0.08 ^a^	0.27 ± 0.02 ^b^	0.33 ± 0.13 ^b,c^	0.48 ± 0.01 ^c,d^	0.59 ± 0.15 ^d^
***Total non-essential amino acids (NEAA)***	286.31 ± 19.25 ^a^	370.75 ± 4.48 ^b^	283.79 ± 20.01 ^a^	357.31 ± 25.58 ^b^	368.43 ± 20.47 ^b^
***ESSENTIAL AMINO ACIDS***					
Histidine	222.76 ± 6.99 ^a^	333.86 ± 3.11 ^b^	221.74 ± 1.69 ^a^	240.24 ± 5.10 ^a^	280.36 ± 22.41 ^c^
Threonine	9.07 ± 0.06 ^a^	11.47 ± 0.96 ^a,b^	11.92 ± 0.71 ^a,b^	13.50 ± 0.26 ^b,c^	16.14 ± 2.23 ^c^
Valine	1.73 ± 0.15 ^a^	4.49 ± 0.18 ^b^	7.48 ± 0.35 ^c^	6.05 ± 0.10 ^d^	8.17 ± 1.09 ^c^
Isoleucine	2.47 ± 0.32 ^a^	3.53 ± 0.54 ^a,b^	4.66 ± 0.14 ^b,c^	4.48 ± 1.30 ^b,c^	5.27 ± 0.73 ^c^
Leucine	4.08 ± 0.32 ^a^	5.26 ± 1.52 ^a^	7.60 ± 0.25 ^b^	5.03 ± 0.05 ^a^	10.57 ± 0.74 ^c^
Phenylalanine	4.36 ± 1.23 ^a,b^	4.17 ± 0.18 ^a^	5.42 ± 0.08 ^a,b,c^	5.57 ± 0.53 ^b,c^	6.10 ± 0.67 ^c^
Lysine	4.48 ± 0.52 ^a^	15.80 ± 0.50 ^b^	16.39 ± 1.76 ^b^	15.96 ± 0.46 ^b^	29.67 ± 0.47 ^c^
Tryptophan	5.24 ± 0.17 ^a^	5.81 ± 0.14 ^a^	7.06 ± 0.35 ^b^	7.16 ± 1.00 ^b^	8.33 ± 0.25 ^c^
***Total essential amino acids (EAA)***	255.53 ± 6.59 ^a^	379.98 ± 2.13 ^b^	274.57± 13.05 ^a^	299.49 ± 2.08 ^a^	346.72 ± 34.79 ^b^
***EAA/NEAA ratio***	0.90 ± 0.08 ^a,b^	1.02 ± 0.01 ^a^	0.95 ± 0.04 ^a,b^	0.84 ± 0.07 ^b^	0.94 ± 0.06 ^a,b^
***Total free amino acids***	541.83 ± 12.66 ^a^	750.73 ± 6.30 ^b^	563.84 ± 37.95 ^a^	656.80 ± 23.50 ^c^	743.10 ± 26.87 ^b,c^

^a–d^ Mean values of at least three determinations ± standard deviation with different superscripts in the same row were significantly different (*p* < 0.05).

**Table 2 foods-09-01377-t002:** Free amino acid content of raw and canned European eels packed in olive oil or spiced olive oil throughout the different steps of the canning process and after 2 and 12 months of storage (expressed as mg/100 g muscle).

	Raw	Frying (in Olive Oil)	Sterilization		Storage 2 Months		Storage 12 Months	
			Olive Oil	Spiced Olive Oil	Olive Oil	Spiced Olive Oil	Olive Oil	Spiced Olive Oil
***NON-ESSENTIAL AMINO ACIDS***								
Aspartic acid	14.24 ± 0.54 ^a^	11.97 ± 2.54 ^a^	37.49 ± 2.01 ^b^	39.39 ± 5.66 ^b^	29.80 ± 1.11 ^c^	32.33 ± 1.38 ^c^	32.26 ± 0.40 ^c^	23.96 ± 0.43 ^d^
Glutamic acid	21.60 ± 1.75 ^a,b^	23.21 ± 0.95 ^a,b^	21.35 ± 0.46 ^a,c^	22.79 ± 4.38 ^a,b^	20.62 ± 1.38 ^a,c,d^	24.02 ± 0.36 ^b^	18.79 ± 0.20 ^c,d^	17.80 ± 1.26 ^d^
Hydroxyproline	0.86 ± 0.04 ^a^	0.82 ± 0.19 ^a^	0.84 ± 0.07 ^a^	0.85 ± 0.20 ^a^	0.81 ± 0.03 ^a^	0.90 ± 0.04 ^a^	1.32 ± 0.14 ^b^	1.77 ± 0.24 ^c^
Serine	3.81 ± 0.03 ^a^	3.33 ± 0.23 ^a^	2.26 ± 0.27 ^b,c^	2.69 ± 0.33 ^b^	1.98 ± 0.01 ^c^	3.38 ± 0.19 ^a^	2.19 ± 0.15 ^b,c^	2.31 ± 0.32 ^b,c^
Glycine	50.24 ± 2.29 ^a^	35.82 ± 3.40 ^b^	48.26 ± 1.24 ^a^	54.17 ± 6.91 ^a^	37.33 ± 2.20 ^b^	50.08 ± 5.24 ^a^	29.85 ± 2.48 ^c^	33.74 ± 2.34 ^b^
Arginine	80.58 ± 6.07 ^a^	100.67 ± 6.97 ^b^	115.08 ± 7.07 ^c^	130.44 ± 3.88 ^d^	137.07 ± 16.04 ^d^	143.58 ± 2.32 ^d^	319.70 ± 5.93 ^e^	247.61 ± 11.62 ^f^
Alanine	12.37 ± 1.13 ^a^	14.29 ± 0.92 ^b^	13.90 ± 0.50 ^a,b^	18.74 ± 0.26 ^c^	13.58 ± 1.36 ^a,b^	17.78 ± 1.02 ^c^	13.57 ± 1.12 ^a,b^	17.84 ± 0.07 ^c^
Proline	2.68 ± 0.28 ^a^	5.99 ± 0.52 ^c,d^	5.31 ± 0.30 ^b,c^	4.82 ± 0.19 ^b^	5.52 ± 0.56 ^b,c^	6.81 ± 0.67 ^e^	5.66 ± 0.32 ^b,c^	6.46 ± 0.57 ^d,e^
Tyrosine	5.98 ± 0.32 ^a^	10.29 ± 0.94 ^c^	10.54 ± 0.30 ^c^	9.73 ± 0.26 ^c^	8.40 ± 0.11 ^b^	9.98 ± 0.20 ^c^	8.55 ± 0.38 ^b^	10.78 ± 0.35 ^c^
Taurine	91.63 ± 6.62 ^a^	119.00 ± 14.70 ^b^	103.84 ± 6.23 ^a,b^	100.06 ± 23.40 ^a,b^	76.71 ± 11.62 ^c^	100.57 ± 14.61 ^a,b^	66.77 ± 6.73 ^c^	60.23 ± 1.73 ^c^
Ornithine	1.58 ± 0.08 ^a^	0.33 ± 0.03 ^b^	0.73 ± 0.12 ^c^	0.48 ± 0.15 ^b,d^	0.65 ± 0.03 ^c,d^	0.61 ± 0.15 ^c,d^	0.54 ± 0.06 ^d^	0.43 ± 0.05 ^b,c^
***Total non-essential amino acids (NEAA)***	286.31 ± 19.25 ^a^	325.14 ± 21.72 ^a,b^	346.70 ± 35.87 ^b^	373.26 ± 20.80 ^b,d^	332.48 ± 4.31 ^a,b^	357.76 ± 30.75 ^b^	499.09 ± 14.42 ^c^	408.42 ± 28.36 ^d^
***ESSENTIAL AMINO ACIDS***								
Histidine	222.76 ± 6.99 ^a^	225.23 ± 13.63 ^a,c^	264.47 ± 3.01 ^b^	270.09 ± 5.48 ^b^	279.62 ± 4.74 ^b^	259.97 ± 1.19 ^b,c^	96.36 ± 8.98 ^c^	139.64 ± 7.28 ^e^
Threonine	9.07 ± 0.06 ^a^	9.61 ± 1.42 ^a^	13.85 ± 1.29 ^b^	13.91 ± 0.15 ^b^	14.12 ± 0.53 ^b^	12.51 ± 0.39 ^b^	12.71 ± 0.55 ^b^	12.22 ± 1.08 ^b^
Valine	1.73 ± 0.15 ^a^	4.40 ± 0.71 ^b^	10.81 ± 0.73 ^d^	7.46 ± 0.48 ^c^	6.64 ± 0.05 ^c^	7.40 ± 0.35 ^c^	26.73 ± 4.01 ^e^	11.26 ± 1.47 ^d^
Isoleucine	2.47 ± 0.32 ^a^	4.36 ± 0.50 ^b,c,e^	3.91 ± 0.47 ^b,c,d^	3.77 ± 0.19 ^b,d^	3.09 ± 0.02 ^a,d^	4.65 ± 0.41 ^c,e^	3.90 ± 0.36 ^c,b,d^	4.68 ± 0.63 ^e^
Leucine	4.08 ± 0.32 ^a^	6.20 ± 0.59 ^b^	6.83 ± 0.33 ^b,c^	6.70 ± 0.55 ^b^	4.27 ± 0.02 ^a^	7.67 ± 0.01 ^c^	4.77 ± 0.08 ^a^	6.05 ± 0.10 ^b^
Phenylalanine	4.83 ± 1.18 ^a,b,c^	5.36 ± 0.94 ^c^	3.84 ± 0.98 ^a,b^	5.36 ± 1.06 ^c,d^	3.69 ± 0.24 ^a,b^	6.60 ± 0.28 ^d^	3.59 ± 0.10 ^a^	5.11 ± 0.52 ^b,c^
Lysine	4.48 ± 0.52 ^a^	23.33 ± 2.53 ^b^	23.32 ± 2.70 ^b^	24.89 ± 1.35 ^b,d^	22.55 ± 2.15 ^b^	28.35 ± 1.36 ^d^	16.79 ± 0.85 ^c^	17.17 ± 0.72 ^c^
Tryptophan	5.24 ± 0.17 ^a^	6.90 ± 0.38 ^b^	7.06 ± 0.43 ^b^	7.15 ± 1.06 ^b^	6.91 ± 0.17 ^b^	7.48 ± 0.66 ^b^	7.77 ± 0.17 ^b^	6.95 ± 1.00 ^b^
***Total essential amino acids (EAA)***	255.53 ± 6.59 ^a^	288.41 ± 17.03 ^b^	335.50 ± 0.58 ^c^	326.25 ± 13.43 ^c^	340.90 ± 7.44 ^c^	316.30 ± 12.03 ^c^	174.83 ± 3.23 ^d^	186.71 ± 8.21 ^d^
***EAA/NEAA ratio***	0.90 ± 0.08 ^a^	0.95 ± 0.17 ^a^	0.94 ± 0.04 ^a^	0.94 ± 0.13 ^a^	1.03 ± 0.04 ^a^	0.90 ± 0.20 ^a^	0.35 ± 0.04 ^b^	0.44 ± 0.03 ^b^
***Total free amino acids***	541.83 ± 12.66 ^a^	601.49 ± 25.99 ^b^	702.91 ± 0.71 ^c^	699.52 ± 7.37 ^c^	673.38 ± 3.13 ^c,d^	713.84 ± 10.36 ^c^	661.21 ± 36.39 ^d^	611.12 ± 0.45 ^b^

^a–f^ Mean values of at least three determinations ± standard deviation with different superscripts in the same row were significantly different (*p* < 0.05).

**Table 3 foods-09-01377-t003:** Biogenic amines (mg/kg muscle) and biogenic amine index of raw and canned European eels in each step during the canning process and at 2 and 12 months of storage, in sunflower oil, in olive oil and spice olive oil.

		Biogenic Amine Content						Total Biogenic Amines	Biogenic Amine Index
		Histamine	β-Phenylethylamine	Tyramine	Putrecine	Cadaverine	Spermidine		
**Raw**		ND	3.10 ± 0.73 ^a^	1.61 ± 0.07 ^a^	0.46 ± 0.06 ^a^	1.40 ± 0.06 ^a^	2.36 ± 0.17 ^a,f^	8.91 ± 0.46 ^a^	3.50 ± 0.36 ^a^
**Salting**		ND	3.05 ± 0.72 ^a^	5.69 ± 0.95 ^b^	1.89 ± 0.29 ^b^	4.73 ± 0.29 ^b^	5.33 ± 0.22 ^b^	20.67 ± 1.69 ^b^	12.30 ±1.18 ^b^
**Sunflower oil**	**F**	ND	4.54 ± 0.25 ^b,c,f^	4.98 ± 0.50 ^b^	1.28 ± 0.27 ^c^	4.59 ± 0.33 ^b^	2.90 ± 0.85 ^a,d^	18.59 ± 1.49 ^c^	10.99 ± 0.72 ^c^
	**S**	ND	4.01 ± 0.21 ^c,g^	1.90 ± 0.58 ^a^	0.86 ± 0.19 ^d,f^	4.10 ± 0.19 ^b^	2.68 ± 0.32 ^a,f^	13.87 ± 0.52 ^e^	7.49 ±0.45 ^d^
	**ST2**	ND	6.13 ± 0.11 ^d^	1.15 ± 0.04 ^c^	1.00 ± 0.22 ^d^	4.60 ± 0.22 ^b^	2.71 ± 0.12 ^a^	15.66 ± 0.08 ^f^	6.86 ± 0.25 ^d^
	**ST12**	ND	8.23 ± 0.55 ^e^	ND	0.62 ± 0.07 ^e^	1.36 ± 0.10 ^a^	0.34 ± 0.04 ^c^	10.52 ± 0.79 ^g,h^	2.12 ± 0.06 ^e^
**Olive oil**	**F**	ND	4.97 ± 0.30 ^f^	1.41 ± 0.21 ^a,d^	0.88 ± 0.13 ^d,f^	4.32 ± 0.16 ^b^	3.60 ± 0.67 ^a,d^	14.51 ± 0.57 ^e,f^	6.64 ± 0.73 ^d^
	**S**	ND	3.64 ±0.44 ^a,c^	1.60 ± 0.17 ^a^	0.69 ± 0.15 ^e,f^	4.30 ± 0.15 ^b^	2.76 ± 0.42 ^a,d^	12.55 ± 0.34 ^d^	6.63 ± 0.68 ^d^
	**ST2**	ND	3.56 ± 0.08 ^a,g^	1.81 ± 0.03 ^a^	0.63 ± 0.04 ^e^	5.51 ± 0.06 ^c^	3.14 ± 0.03 ^d^	15.04 ± 1.19 ^e,f^	7.22 ± 0.31 ^d^
	**ST12**	ND	4.35 ± 0.17 ^b,c^	0.83 ± 0.03 ^e^	0.42 ± 0.04 ^a^	6.28 ± 0.31 ^c^	1.79 ± 0.18 ^e,f^	14.13 ± 0.28 ^e^	7.51 ± 0.35 ^d^
**Spiced olive oil**	**S**	ND	3.02 ± 0.12 ^a^	1.33 ± 0.07 ^a,d^	0.63 ± 0.01 ^e^	4.64 ± 0.02 ^b^	2.16 ± 0.22 ^f^	11.69 ± 0.57 ^d,g^	6.60 ± 0.37 ^d^
	**ST2**	ND	3.21 ± 0.55 ^a^	1.80 ± 0.08 ^a^	1.09 ± 0.03 ^d^	5.76 ± 0.04 ^c^	3.37 ± 0.17 ^d^	15.04 ± 0.73 ^e,f^	8.66 ± 0.02 ^f^
	**ST12**	ND	7.19 ± 0.54 ^h^	ND	0.30 ± 0.01 ^g^	1.93 ± 0.01 ^e^	0.31 ± 0.03 ^c^	9.99 ± 0.13 ^a,h^	2.28 ± 0.67 ^e^

^a–h^ Means in the same column with different letters differ significantly (*p* < 0.05). F: frying, S: sterilization process; ST2: storage 2 months; ST12: storage 12 months. ND: not detected.

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
