# Peer review of "Free Amino Acids and Biogenic Amines in Canned European Eels: Influence of Processing Step, Filling Medium and Storage Time"

_foods, 2020, doi:10.3390/foods9101377_

Round 1

Reviewer 1 Report

Presented article is well written and contains the results of the canning process and different filling media on the free  amino  acid  and  biogenic  amine  contents  of characterization of canned European eels. The findings provide a better understanding of the canning process and filling medium on free amino acids and biogenic amines composition.

I hove only two comments:

Authors stated that they performed variance. Please specify it was one-way, two-way or three-way ANOVA?

The significant differences between means should be marked on Fig. 1.

Author Response

Reviewer #1:

First of all, we wish to thank the reviewer for his work and dedication to our article. His useful comments and suggestions really contributed to the improvement of our manuscript.

The modifications made in the revised version as a consequence of the Reviewer 1’s suggestions were highlighted with "Tracked Changes"

Authors stated that they performed variance. Please specify it was one-way, two-way or three-way ANOVA?

 Following the reviewer’s suggestion, we have added the ANOVA type. Line 178

The significant differences between means should be marked on Fig.

Following the reviewer’s suggestion, we have added the significant differences on Fig.1. Lines 200-201

Reviewer 2 Report

In my opinion the subject of the manuscript is very interesting and well described.

The introduction and research methodology are well edited and written, I have no objections to these parts of the manuscript.

The results are clearly presented, but some of them require more detailed discussion with the literature data.

Please discuss changes in moisture content during storage the canned eels packed in sunflower oil and in spiced olive oil. The importance of these changes for FAA content and biogenic amine formation should also be discussed.

In my opinion, on figure 1, the significance of the differences between the presented results should be presented.

Please explain and discuss the differences in FAA in canned eels packed in olive oil and canned eels packed in sunflower oil. What is the reason for the differences between these two canned eels? Could it be due to the fatty acid composition of these oils? Did it have an effect on the biogenic amine content and biogenic amine index?

Author Response

Reviewer #2:

First of all, we wish to thank the reviewer for his work and dedication to our article. His useful comments and suggestions really contributed to the improvement of our manuscript.

The modifications made in the revised version as a consequence of the Reviewer 2’s suggestions were highlighted with "Tracked Changes"

The results are clearly presented, but some of them require more detailed discussion with the literature data.

Please discuss changes in moisture content during storage the canned eels packed in sunflower oil and in spiced olive oil. The importance of these changes for FAA content and biogenic amine formation should also be discussed.

Following the reviewer’s suggestion, we have added some more information about these changes.

In my opinion, on figure 1, the significance of the differences between the presented results should be presented.

Following the reviewer’s suggestion, we have added the significant differences on Fig.1. Lines 200-201

Please explain and discuss the differences in FAA in canned eels packed in olive oil and canned eels packed in sunflower oil. What is the reason for the differences between these two canned eels? Could it be due to the fatty acid composition of these oils? Did it have an effect on the biogenic amine content and biogenic amine index?

Following the reviewer’s suggestion, we have added some more information. Lines 316-320

Reviewer 3 Report

Dear Editor,

in my opinion, the manuscript shows interesting results about the effects of the canning process and different filling media on the free amino acid and biogenic amine contents of eels.

The manuscript is well written and can be read well. There are only a few formatting problems in Table 1: some superscript letters have an underline. Why?

In table 3 does not read well "Spice olive oil".

In conclusion, the manuscript can be accepted for publication after tables 1 and 3 have been arranged.

Author Response

Reviewer #3:

First of all, we wish to thank the reviewer for his work and dedication to our article. His useful comments and suggestions really contributed to the improvement of our manuscript.

The modifications made in the revised version as a consequence of the Reviewer 3’s suggestions were highlighted with "Tracked Changes"

In my opinion, the manuscript shows interesting results about the effects of the canning process and different filling media on the free amino acid and biogenic amine contents of eels.

The manuscript is well written and can be read well. There are only a few formatting problems in Table 1: some superscript letters have an underline. Why?

We have reviewed the table 1 and verified that there were no underlines

In table 3 does not read well "Spice olive oil".

The reviewer is right.We have reviewed the table 3 and corrected

In conclusion, the manuscript can be accepted for publication after tables 1 and 3 have been arranged.

Reviewer 4 Report

This manuscript reported the evaluation of the content of free amino acids and biogenic amines in canned eels, with a focus on the level change under the different food processing method, including switching the type of oil, storage time, etc. Several previous researches have been performed to analyze the FAA and biogenic amines from the fish, but not the eel, therefore, the selling point of this manuscript resided in 1) Material; 2) the effect of filling media on the FAA levels, together with the statistical analysis. The results seems promising, and it would definitely attract the attention of readers in this area, therefore, as far as I am concerned, this manuscript can be assigned as minor revision.

Here is my questions and concerns.

  1. Page 5, line 192, as it is stated ”sterilization caused a decrease in the moisture content of canned eels packed in spiced olive oil”. What is the potential reason to explain this? It is highly recommended to add some discussion here.

  1. Similarly, three different types of oil were used in the study to compare the level of FAA, and it seems that different oil has different impact on the FAA content, as list in table 2. For example, the arginine content varied a lot filled with olive oil and spicy olive oil. Why this happened? We understand the data displayed as it is, but it will be great to add the possible answer to explain it.

  1. Again, for the biogenic amines, it was observed that the phenylethylamine content increases were much slower in olive oil than the sunflower oil. Why?

Overall, the results are pretty detailed, but the discussion seems to be less detailed.

Author Response

Reviewer #4:

First of all, we wish to thank the reviewer for his work and dedication to our article. His useful comments and suggestions really contributed to the improvement of our manuscript.

The modifications made in the revised version as a consequence of the Reviewer 4’s suggestions were highlighted with "Tracked Changes"

Page 5, line 192, as it is stated”sterilization caused a decrease in the moisture content of canned eels packed in spiced olive oil”. What is the potential reason to explain this? It is highly recommended to add some discussion here.

Following the reviewer’s suggestion, we have added some more information. Lines 197-201

Similarly, three different types of oil were used in the study to compare the level of FAA, and it seems that different oil has different impact on the FAA content, as list in table 2. For example, the arginine content varied a lot filled with olive oil and spicy olive oil. Why this happened? We understand the data displayed as it is, but it will be great to add the possible answer to explain it.

Following the reviewer’s suggestion, we have added some more information. Lines 318-322

Again, for the biogenic amines, it was observed that the phenylethylamine content increases were much slower in olive oil than the sunflower oil. Why?

Following the reviewer’s suggestion, we have added some more information. Lines 390-392

Overall, the results are pretty detailed, but the discussion seems to be less detailed.

We hope that with the new information the discussion is more complete